# Preparation and Characterization of Nano-Selenium Decorated by Chondroitin Sulfate Derived from Shark Cartilage and Investigation on Its Antioxidant Activity

**DOI:** 10.3390/md20030172

**Published:** 2022-02-26

**Authors:** Jianping Chen, Xuehua Chen, Jiarui Li, Baozhen Luo, Tugui Fan, Rui Li, Xiaofei Liu, Bingbing Song, Xuejing Jia, Saiyi Zhong

**Affiliations:** 1Guangdong Provincial Key Laboratory of Aquatic Product Processing and Safety, Guangdong Provincial Engineering Technology Research Center of Seafood, Guangdong Province Engineering Laboratory for Marine Biological Products, Key Laboratory of Advanced Processing of Aquatic Product of Guangdong Higher Education Institution, Guangdong Provincial Science and Technology Innovation Center for Subtropical Fruit and Vegetable Processing, College of Food Science and Technology, Guangdong Ocean University, Zhanjiang 524088, China; 18977740415@163.com (X.C.); rain8413@163.com (J.L.); 13413650541@163.com (B.L.); ftg2693982106@163.com (T.F.); liruihn@163.com (R.L.); liuxf169@126.com (X.L.); 15891793858@163.com (B.S.); jiaxj@gdou.edu.cn (X.J.); 2Collaborative Innovation Center of Seafood Deep Processing, Dalian Polytechnic University, Dalian 116034, China

**Keywords:** nanoselenium, chondroitin sulfate derived from shark cartilage, structure characterization, antioxidant activity

## Abstract

In the present study, a selenium-chondroitin sulfate (SeCS) was synthesized by the sodium selenite (Na_2_SeO_3_) and ascorbic acid (Vc) redox reaction using chondroitin sulfate derived from shark cartilage as a template, and characterized by SEM, SEM-EDS, FTIR and XRD. Meanwhile, its stability was investigated at different conditions of pH and temperatures. Besides, its antioxidant activity was further determined by the DPPH and ABTS assays. The results showed the SeCS with the smallest particle size of 131.3 ± 4.4 nm and selenium content of 33.18% was obtained under the optimal condition (CS concentration of 0.1 mg/mL, mass ratio of Na_2_SeO_3_ to Vc of 1:8, the reaction time of 3 h, and the reaction temperature of 25 °C). SEM image showed the SeCS was an individual and spherical nanostructure and its structure was evidenced by FTIR and XRD. Meanwhile, SeCS remained stable at an alkaline pH and possessed good storage stability at 4 °C for 28 days. The results on scavenging free radical levels showed that SeCS exhibited significantly higher antioxidant activity than SeNPs and CS, indicating that SeCS had a potential antioxidant effect.

## 1. Introduction

Reactive oxygen species (ROS) are the result of the normal cell metabolism of living organisms. However, when the production of ROS exceeds the scavenging capacity of living organisms, oxidative stress occurs [1]. Excessive reactive oxygen species (ROS) not only affect normal metabolism, but also is closely related to the occurrence and development of various diseases, including cardiovascular diseases [2], Alzheimer’s disease [3], nonalcoholic fatty liver disease [4], and so on. As an essential micronutrient for human health, selenium (Se) plays a vital role in protecting against oxidative stress [5]. Nevertheless, the beneficial and harmful doses of Se is an extremely narrow margin, which limits its practical applications in food and medicine [6]. Therefore, it is necessary to take effective measures to solve this problem. 

Nanotechnology, as a technology developed in recent years, has been widely used in the field of biology. Selenium nanoparticles (SeNPs) have been widely used as drug carriers due to their advantages of small volume, large specific surface area, and unique physical and chemical properties. Previous studies have found that SeNPs can remove harmful peroxides from the body through glutathione peroxidase (GSH-PX) and protect the membrane structure of the body from damage [7]. Meanwhile, compared with organic and inorganic selenium compounds [8], SeNPs possesses higher bioactivity like antioxidant activity, better bioavailability and lower toxicity [9]. However, SeNPs are very unstable in the liquid phase and tend to aggregate and form gray or black selenium with a large particle size, thus losing the bioavailability and bioactivity of SeNPs [10]. Therefore, a suitable stabilizer is needed to improve its stability.

Natural bioactive polysaccharides have attracted more and more attention because they are rich in hydrophilic groups, such as hydroxyls, and are considered as ideal templates for SeNPs stabilization. A variety of polysaccharides have been reported to modify SeNPs and the modified SeNPs have been found to have strong antioxidant activity [6,11,12]. Chondroitin sulfate (CS), a natural anionic glycoaminoglycan, is extracted from various marine animal cartilages, such as shark, skate and so on [13,14]. Studies showed CS exhibited a variety of biological activities, such as antioxidant activity [15], anti-inflammatory activity [16] and anti-tumor activity [17]. Since it is rich in carboxyl and hydroxyl groups, it is often used to stabilize SeNPs. For example, Han et al. reported the preparation method of selenium–chondroitin sulfate (SeCS) using CS as a stabilizer to stable SeNPs, but the Se content in SeCS is only 10.1%, and the study on its stability has not been reported [18]. Since the Se content and stability of SeCS are related to its biological activity, it is necessary to further improve the Se content and investigate the stability of SeCS. Therefore, in this study, we screened the preparation conditions of SeCS, investigated its stability at a different pH and temperature, and evaluated its antioxidant activity using 2,2-diphenyl-1-picrylhydrazyl (DPPH) and 2,2′-azino-bis (3-ethylbenzo-thiazoline-6-sulfonic acid) diammonium salt (ABTS) assays. This study provides a theoretical basis for the application of SeCS as a fresh antioxidant agent in food and medicine. 

## 2. Results and Discussion

### 2.1. Determination of Preparation Conditions of SeCS

#### 2.1.1. Determination of CS Concentration

In order to evaluate the effect of CS concentration on the formation of SeCS, the particle sizes of SeCS prepared with different concentrations of CS were measured by nanoparticle size analyzer. As shown in Figure 1A, the particle sizes of SeNPs with 0.05 mg/mL CS surface decoration were 278.70 ± 9.47 nm. The average particle diameters of SeNPs were significantly decreased to 113.7 ± 2.89 nm in the presence of 0.1 mg/mL CS, indicating that CS between 0.05 mg/mL and 0.1 mg/mL could inhibit the aggregation of SeNPs with the increase of CS concentration. The result was in accordance with the previous study, which demonstrated that the particle diameters of SeNPs decorated with 1.0 mg/mL chitosan were smaller than that synthesized with 0.6 mg/mL chitosan [19]. However, when CS concentration is greater than 0.1 mg/mL, the particle size of SeCS in the system became larger. This might be because CS molecules aggregated to reduce its hydrophilic groups’ function, resulting in the particle size become larger. Therefore, the optimal concentration of CS was selected as 0.1 mg/mL.

#### 2.1.2. Determination of Molar Ratio of Na_2_SeO_3_ to Vc

According to the reaction equation, sodium selenite (Na_2_SeO_3_) reacts with ascorbic acid (Vc) at the coefficient of 1:2. However, in fact, an excess of V_C_ needs to be added to the system to prevent SeNPs from being oxidized. Therefore, in this study, fixing the CS concentration of 0.1 mg/mL, the reaction time of 3 h and the reaction temperature of 55 °C, the molar ratio of Na_2_SeO_3_ to Vc varied from 1:2 to 1:10. The particle sizes of SeCS prepared with different molar ratios of Na_2_SeO_3_ to Vc were shown in Figure 1B. The results showed that the average particle diameters of SeCS obtained with the molar ratio of Na_2_SeO_3_ to Vc of 1:2 were 187.3 ± 12.48 nm. As the molar ratio of Na_2_SeO_3_ to Vc increased to 1:8, the particle size reached the smallest. It was speculated that excessive Vc could improve the growth of the crystal nucleus to stabilize SeCS [20]. However, further increasing Vc concentration meant the particle size became larger due to the unstable reaction system. So, it might be the reason why the diameter of the SeCS prepared with a molar ratio of Na_2_SeO_3_ to Vc of 1:10 was larger than that obtained with a molar ratio of Na_2_SeO_3_ to Vc of 1:8. Therefore, the optimal molar ratio of Na_2_SeO_3_ to Vc was selected as 1:8. 

#### 2.1.3. Determination of Reaction Time 

Studies showed that reaction time had a vital role to regulate the SeNPs reaction system [21]. If the reaction time is too short, the reaction between Na_2_SeO_3_ and Vc is not sufficient. However, if the reaction time is too long, the SeNPs in the system will gather, leading to increasing the particle size. Therefore, we next further investigated the influence of reaction time on the particle size of system. Fixing the CS concentration of 0.1 mg/mL, the molar mass ratio of Na_2_SeO_3_ to Vc of 1:8 and the reaction temperature of 55 °C, the reaction time varied from 1 h to 4 h. The result was shown in Figure 1C. When the reaction time extended from 1 h to 2 h, no significant difference was observed in the particle size of the obtained SeCS. When the reaction time increased to 3 h, the particle size of the obtained SeCS reached to 131.2 ± 9.97 nm. When the reaction time exceeded 3 h, the particle size of the obtained SeCS remarkably increased. Therefore, we selected the reaction time of 3 h as the optimal reaction time to prepare SeCS. 

#### 2.1.4. Determination of Reaction Temperature 

Fixing the CS concentration of 0.1 mg/mL, the molar mass ratio of Na_2_SeO_3_ to Vc of 1:8, reaction time of 3 h, the reaction temperature varied from 25 to 85 °C. The particle size of the obtained SeCS is shown in Figure 1D. It could be seen that although the particle size of the obtained SeCS slightly decreased at 55 °C, there was no significant difference between 25 °C and 55 °C. When the reaction temperature was further separately increased to 70 °C and 85 °C, the particle size of the obtained SeCS both significantly increased. The reason might be that heating easily led to the violent movement of nanoparticles, thus increasing the chance and intensity of collisions and intensifying aggregation [22]. Therefore, in terms of saving energy, the optimal reaction temperature was selected as 25 °C. 

In summary, these above results demonstrated that we optimized conditions for preparing SeCS as a CS concentration of 0.1 mg/mL, the molar mass ratio of Na_2_SeO_3_ to Vc of 1:8, reaction time of 3 h and reaction temperature of 25 °C, so SeCS used in subsequent experiments were obtained in optimal reaction conditions. Under the optimal reaction conditions, the SeCS was successfully prepared and inductively coupled plasma mass spectrometry (ICP-MS) was used to measure the Se content of the SeCS obtained. It was found that the Se content in SeCS was 33.18%, which is higher than that reported in previous studies [18,23].

### 2.2. Characterization of SeCS 

#### 2.2.1. Scanning Electron Microscopy (SEM) 

In order to verify the above results, SEM was used to analyze the morphology of SeNPs and SeCS. As shown in Figure 2, the SEM results of SeNPs clearly revealed that SeNPs without CS surface decoration showed serious aggregation and formed large particles chunks (Figure 2A,C). Nevertheless, the addition of CS accelerated the production of homogeneous spherical SeCS with high dispersibility (Figure 2B,D), which further confirmed the formation of SeCS.

#### 2.2.2. Element Analysis of SeCS

To determine chemical compositions of SeCS samples, scanning electron microscopy equipped with an energy dispersion spectrum detector (SEM-EDS) was employed. The results were shown in Figure 3. It was found that C (44.16%), O (37.74%), and Na (18.1%) were the chemical compositions of CS (Figure 3A). Furthermore, an SEM-EDS investigation of SeNPs showed the percentages of Se, C and O atoms were 92.27%, 7.43% and 0.30% (Figure 3B), which were similar to previous findings of Ye et al. [21]. An EDS investigation of SeCS (Figure 3C) showed that the presence of a strong Se atoms signal (65.73%). The existence of the Na atom (1.27%), C atom (25.55%) and O atom (7.44%) in SeCS suggested that CS successfully combined to the surface of the SeNPs.

#### 2.2.3. Fourier Transform Infrared Spectroscopy (FT-IR) Analysis

To confirm the chemical binding of CS to the surface of the SeNPs, FTIR spectroscopy is used to ascertain the formation of SeCS. The FTIR spectra of CS and SeCS were shown in Figure 4. The typical IR spectrum of CS was presented in Figure 4 which was in good agreement with the literature [23]. The FTIR spectrum of CS exhibited an absorption band at 3402 cm^−1^, indicating the overlapping of –OH and –NH stretching vibrations. Additionally, the absorption peaks of CS were 1070 cm^−1^ and 1128 cm^−1^, 1224 cm^−1^, 1419 cm^−1^, and 1635 cm^−1^, which corresponded to the characteristic asymmetric stretching vibrations of the C–O–C bridge (β-1,4 glycosidic bonds), asymmetric stretching vibrations of –S=O, stretching vibrations of the -COOH and stretching vibrations of the –C=O of –NHCO–. By comparing the FTIR spectra of CS, SeCS resembled that of CS and there were no new absorption peaks in the FTIR spectrum of SeCS, indicating that the reaction between CS and SeNPs did not generate any new covalent bonds. However, an obvious change occurred in the peak locations of SeCS, indicating that the main interaction between CS and SeNPs was physical adsorption. 

#### 2.2.4. Powder X-ray Diffractometry (XRD) Analysis

On this basis of FT-IR analysis, XRD was used to characterize the formation of SeCS. XRD analysis could detect phase identification of crystalline materials. The intensity and sharpness of XRD peaks reflected the crystalline nature of the sample. The XRD spectra of CS, SeNPs and SeCS were shown in Figure 5. As shown in Figure 5, the X-ray diffractogram of SeNPs showed two broad peaks in the ranges of 20–40° and 40–60° (2θ), indicating that SeNPs existed in an amorphous form. This result was in good agreement with the literature [21]. No sharp peaks were found in the XRD pattern of CS and SeCS, confirming their amorphous characteristic in nature. Compared with the XRD diffraction patterns of CS and SeNPs, both the peak positions and intensity of SeCS shifted, suggesting that the formation of SeCS. 

#### 2.2.5. Stability of SeCS 

The stability of nanoparticles is one of the key factors for their biological function. Studies have shown that the stability of nanoparticles is closely related to pH, storage temperature and storage time in the application medium [24,25]. Therefore, we next investigated the influence of different pHs and storage temperatures on the stability of SeCS. The influence of pH on the stability of SeCS was shown in Figure 6A. The particle diameters of SeCS notably decreased to approximately 98.9 ± 5.9 nm at the 2–8 pH range. Then, no significant shift occurred at the 8–12 pH range. This was probably attributed to the strongest electrostatic interaction between anionic CS and SeNPs at pH 8.0 due to the sensitivity of CS at a low pH [26]. The effect of storage temperature on the stability of SeCS was shown in Figure 6B. When the SeCS solution was stored at 4 °C for 28 days, the particle size showed no obvious change. Yet, the particle diameter of the SeCS solution stored at 25 °C for 28 days significantly increased to 262.7 ± 13.6 nm. It was speculated that an increase in particle diameter might be related with the changes of the internal structure of the SeCS because the increasing temperature resulted in a change in the amorphous state of SeCS to the crystalline state. Moreover, our results were consistent with previous studies that high temperature was not conducive to the stability of selenium nanoparticles [10,26]. These results indicated that SeCS exerted excellent stability under a refrigerating temperature and alkaline environment. 

### 2.3. The Antioxidant Property of SeCS

The antioxidant potential of SeCS was analyzed using the DPPH assay and ABTS assay in which ascorbic acid was used as a standard. As shown in Figure 7, the DPPH and ABTS radical scavenging rates of CS and SeNPs were extremely low. However, comparing with CS and SeNPs, the DPPH and ABTS radical scavenging rates of SeCS significantly increased. With the concentration of SeCS increased, the DPPH and ABTS radical scavenging rates of SeCS increased from 29.13 ± 3.28% (0.1 mg/mL) and 13.92 ± 2.57% (20 μg/mL) to 66.69 ± 2.71% (0.5 mg/mL) and 52.44 ± 2.29% (100 μg/mL), respectively. Our results indicated that SeCS could effectively scavenge the DPPH and ABTS free radical in a dose-dependent manner. However, the DPPH and ABTS scavenging activities of SeCS were lower than the V_C_, particularly in the ABTS scavenging assay.

## 3. Materials and Methods

### 3.1. Materials 

Chondroitin sulfate (CS) derived from shark cartilage (95% purity, Mw = 499.37, C107703) was purchased from Aladdin. CS is composed of chondroitin 6-sulfate and chondroitin 4-sulfate, and the proportion of chondroitin 6-sulfate and chondroitin 4-sulfate detected by high performance liquid chromatography is equal to or greater than 0.33:1. Na_2_SeO_3_, ascorbic acid (Vc), DPPH and ABTS were supplied by Shanghai Yuanye Biotechnology Co., Ltd. (Shanghai, China). The other chemicals and reagents were of analytical purity grade. 

### 3.2. Determination of Preparation Conditions of SeCS Nanoparticles 

SeCS nanoparticles were prepared according to the method as described in previous study with some modifications [23]. CS was dissolved in 30 mL of deionized water and the concentration of CS was 0.05, 0.1, 0.15, 0.2, and 0.25 mg/mL. Different molar ratios of Na_2_SeO_3_ to Vc (Na_2_SeO_3_: Vc = 1:2, 1:4, 1:6, 1:8, 1: 10) were added to CS solution and stirred at different temperatures (25 °C, 40 °C, 55 °C, 70 °C, 85 °C) for different times (1 h, 2 h, 3 h, 4 h). Then, the solution was dialyzed against double-distilled water for 48 h with a dialysis bag (M_W_ = 3500). The optimum preparation process of SeCS was determined by particle size analysis. The final solution was freeze-dried to preserve the bioactivity of SeCS.

### 3.3. Measurement of Se Content in SeCS

The selenium content of SeCS was measured using 7500 CX ICP-MS (Agilent, Palo Alto, CA, USA). The following measurement conditions of the instrument: incident power of 1550 W; plasma gas flow rate at 15 L/min; carrier gas flow rate at 1.0 L/min; auxiliary gas flow rate at 1.0 L/min; helium flow rate at 4.0 mL/min; atomization chamber temperature of 2 °C; sampling depth of 10.0 mm; sampling rate at 1.0 L/min. 

### 3.4. Characterization of SeCS

The surface morphologies of the samples were detected using MIRA4 SEM (Tescan, Brno, Czech). The elemental compositions of the samples were measured using a MIRAL SEM-EDS (Tescan, Brno, Czech). The FT–IR spectrums of the samples were measured using a Tensor 27 spectrometer (Bruker, Karlsruhe, Germany) from 4000 to 400 cm^−1^ with a 4 cm^−1^ resolution, i.e., 2 mg of the sample was completely ground with the spectroscopic grade potassium bromide (KBr) powder and a transparent tablet was used for measurement. The X-ray diffractometer of the samples was obtained using an Ultima VI diffractometer (Rigaku, Tokyo, Japan) operated at 40 kV and 40 mA. Moreover, the samples were investigated in the range of 10–80 degree (2θ angle). 

### 3.5. Stability of SeCS in Different Environments

#### 3.5.1. Effects of Different pH on the Size of SeCS Nanoparticles

After dialysis, the system solution pH (6.8) was adjusted to 2, 4, 6, 8, 10 and 12 using 0.1 M sodium hydroxide (NaOH) or 0.1 M hydrochloric acid (HCl). After 10 min, the particle size of SeCS was determined by Malvern Zeizer Nano ZS particle analyzer (Malvern, Worcestershire, UK).

#### 3.5.2. Effects of Different Storage Temperatures on the Size of SeCS Nanoparticles

The SeCS solutions were stored in 4 °C and 25 °C, respectively. After 0, 7, 14 and 28 days, the size of SeCS was measured by Malvern Zetasizer Nano ZS particle analyzer (Malvern, Worcestershire, UK).

### 3.6. Antioxidant Activity Evaluation of SeCS

#### 3.6.1. DPPH Radical Scavenging Assay

The antioxidant activity of SeCS was evaluated using 2,2-diphenyl-1-picrylhydrazyl (DPPH) reagent according to a previously described method with some modifications [27]. Different concentrations (0.1–0.5 mg/mL) of CS, SeNPs and SeCS solutions were respectively added into 0.2 mM DPPH solution in methanol. After 30 min incubation in a dark place, the color change of the reaction mixture was read at 517 nm using a TU-1901 double beam UV-visible spectrophotometry (Persee, Beijing, China). Positive control in this assay was ascorbic acid. The formula of the DPPH scavenging rate was listed below:DPPH scavenging rate (%) = [A_0_ − (A_1_ − A_2_)]/A_0_ × 100%

In the equation, A_0_, A_1_ and A_2_ separately represented the absorbance values of control, the samples with DPPH solution and the samples without DPPH solution.

#### 3.6.2. ABTS Radical Scavenging Assay

ABTS radical scavenging activity was detected as reported in previous research with some modifications [28]. Potassium persulfate (2.6 mM) was added into ABTS (7.4 mM) to obtain the ABTS stock solution. Then, the solution was incubated at room temperature in dark for 12 h. Deionized water was used to dilute the solution to prepare the ABTS working solution (0.70 ± 0.02 absorbance at 734 nm). After that, 4 mL ABTS working solution with 1 mL sample were mixed for 6 min in the dark. Subsequently, the absorbance of mixture was recorded at a wavelength of 734 nm using a TU-1901 double beam UV-Visible spectrophotometer (Persee, Beijing, China). Ascorbic acid was selected as a positive control. The formula of the ABTS scavenging rate was listed below:ABTS scavenging rate (%) = (1 − A_i_/A_0_) × 100

In the equation, A_0_ and A_i_ separately indicated the absorbance values of the blank control and the sample.

### 3.7. Statistical Analysis

All assays were repeated three times and the values were indicated as mean ± standard deviation (SD). Statistical analysis was done using SPSS 23.0 software and Origin 2020. The comparison between two groups was analyzed by Two-tailed Student’s *t*-test. *p* value less than 0.05 or 0.01 was considered significantly shift. 

## 4. Conclusions

This study provided an effective strategy to prepare SeCS using CS as a stabilizing agent. Using the reaction system at reaction conditions of a Na_2_SeO_3_:Vc ratio of 1:8, 0.1 mg/mL CS, a 3 h reaction time and a 25 °C reaction temperature could generate spherical SeCS nanoparticles of 131.3 ± 4.4 nm in diameter with a high selenium entrapment efficiency (33.18%). These amorphous SeNPs were suggested to interact with CS via physical adsorption, and exhibited pH stability (pH > 8) and storage stability at 4 °C for 28 days. Moreover, SeCS exerted a stronger in vitro antioxidant capacity than SeNPs and CS. These results confirmed the higher stability and the improved antioxidant properties of SeNPs capped with CS compared to SeNPs. Taken together, SeCSs have the potential to further develop a dietary supplement to apply in the prevention and alleviation of oxidative stress-related diseases. 

## Figures and Tables

**Figure 1 marinedrugs-20-00172-f001:**
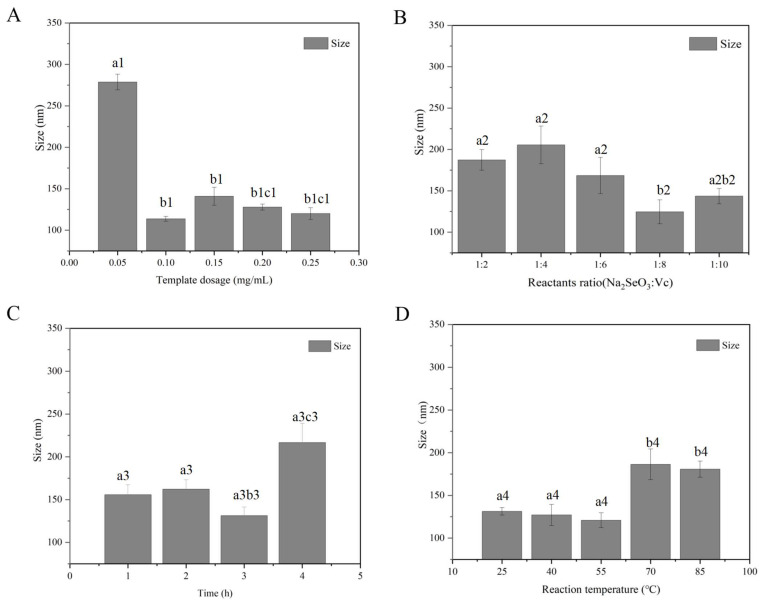
Effects of different template dosage (**A**), reactants ratio (**B**), reaction time (**C**), and reaction temperatures (**D**) on particle size of SeCS. Bars with a1, b1 and c1 represented a statistical difference (*p* < 0.05) among different template dosages. Bars with a2 and b2 represented a statistical difference (*p* < 0.05) among different reactants ratios. Bars with a3, b3 and c3 represented a statistical difference (*p* < 0.05) among different times. Bars with a4 and b4 represented a statistical difference (*p* < 0.05) among different reaction temperatures.

**Figure 2 marinedrugs-20-00172-f002:**
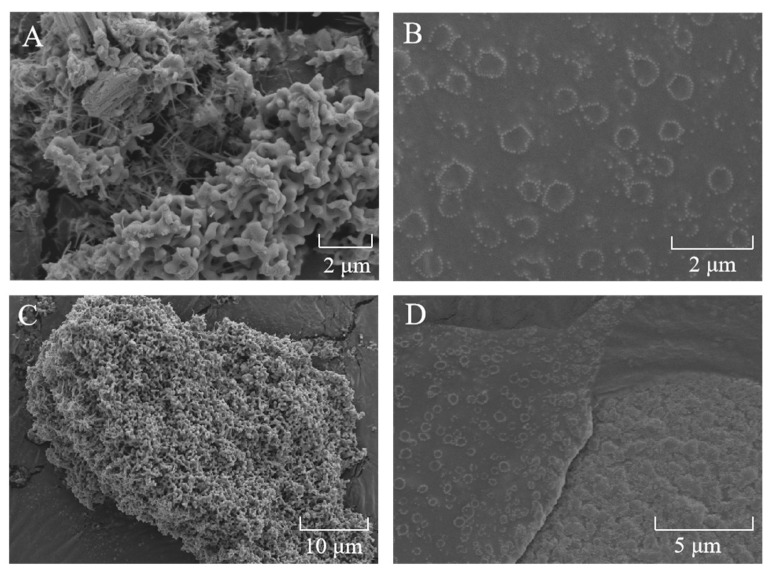
SEM images of SeNPs (**A**,**C**) and SeCS (**B**,**D**) powders. The SeNPs were obtained in the same procedure of SeCS without CS.

**Figure 3 marinedrugs-20-00172-f003:**
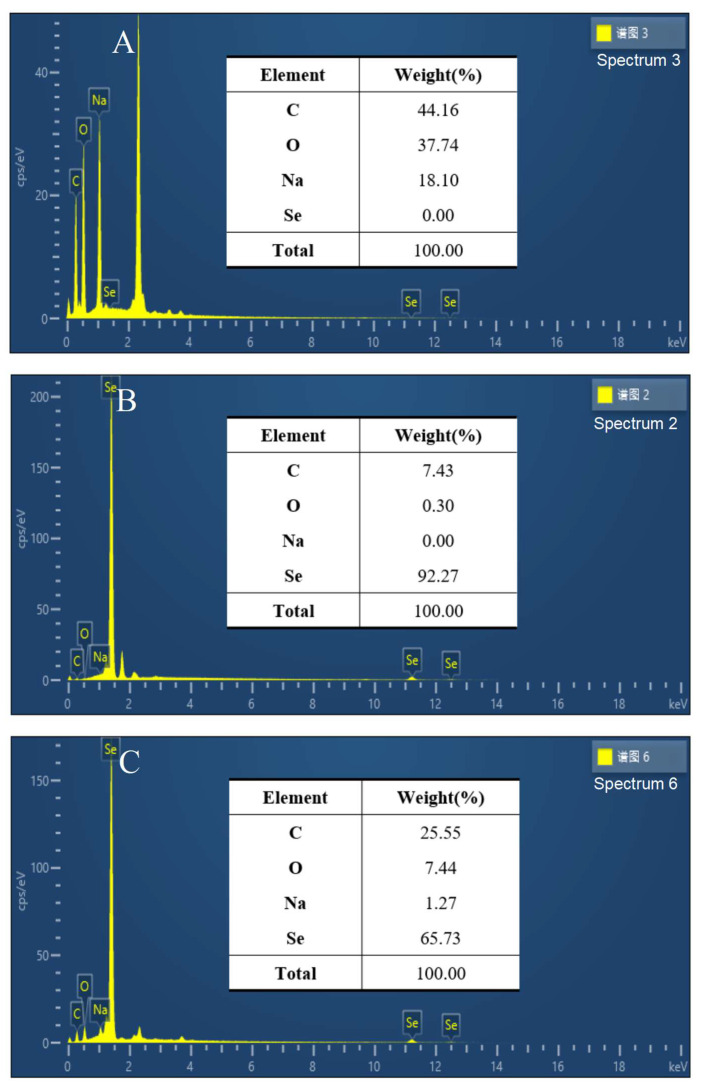
SEM-EDS element analysis of CS (**A**), SeNPs (**B**) and SeCS (**C**). The SeNPs were obtained in the same procedure of SeCS without CS.

**Figure 4 marinedrugs-20-00172-f004:**
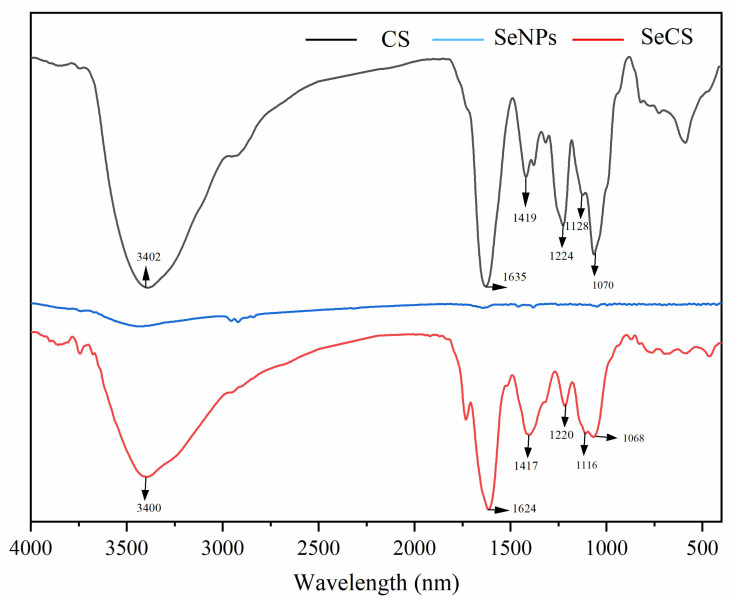
FT-IR spectra of CS, SeNPs and SeCS. The SeNPs were obtained in the same procedure of SeCS without CS.

**Figure 5 marinedrugs-20-00172-f005:**
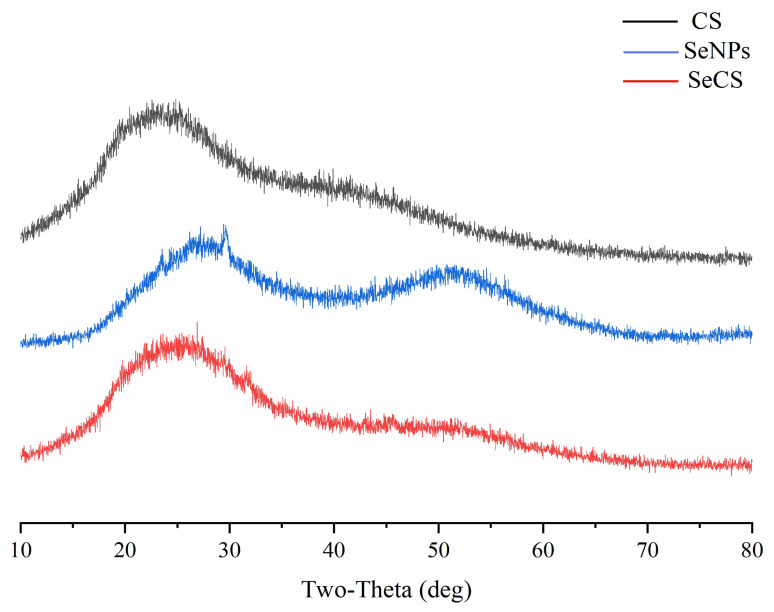
XRD patterns of CS, SeNPs and SeCS. The SeNPs were obtained in the same procedure of SeCS without CS.

**Figure 6 marinedrugs-20-00172-f006:**
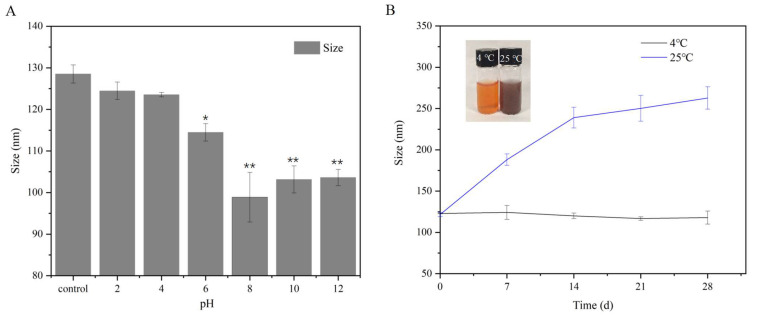
Effects of different pHs (**A**) and storage temperatures (**B**) on the size of SeCS. The initial pH value of the control group was 6.8. *p* < 0.05 (*) or *p* < 0.01 (**) means that columns between control group and other groups are significantly different.

**Figure 7 marinedrugs-20-00172-f007:**
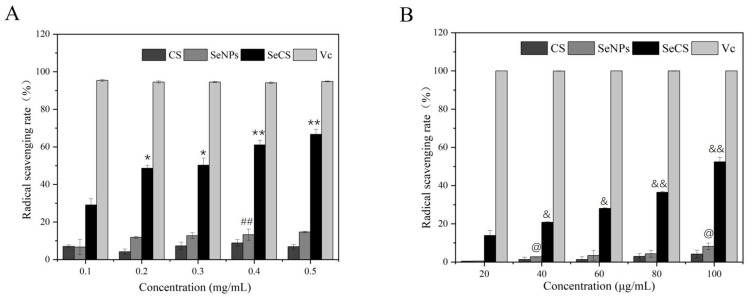
DPPH radical scavenging rate (**A**) and ABTS radical scavenging rate (**B**) of SeCS. *p* < 0.05 (*) or *p* < 0.01 (**) means that columns between SeCS at 0.1 mg/mL and SeCS at other concentrations are significantly different. *p* < 0.01 (##) means that columns between SeNPs at 0.1 mg/mL and SeNPs at 0.4 mg/mL are significantly different. *p* < 0.05 (&) or *p* < 0.01 (&&) means that columns between SeCS at 20 μg/mL and SeCS at other concentrations are significantly different. *p* < 0.05 (@) means that columns between SeNPs at 20 μg/mL and SeNPs at other concentrations are significantly different.

## Data Availability

Not applicable.

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
