# Peer review of "Preparation and Characterization of Nano-Selenium Decorated by Chondroitin Sulfate Derived from Shark Cartilage and Investigation on Its Antioxidant Activity"

_marinedrugs, 2022, doi:10.3390/md20030172_

Round 1
Reviewer 1 Report
Comments to the Authors of manuscript number: marinedrugs-1602809 entitled “Preparation, characterization of selenium–chondroitin sulfate and investigation on its antioxidant activity”.
Authors have presented the process of the formulation of a selenium-chondroitin sulfate, where CS is derived from shark.
- L 19- small letters
- L 41- small letters
- L 69 – all abbreviations should be explained
- L 90 – small letter
- L 90 – why it is sodium selenite? Sodium ion keep a water molecule, and cannot be consumed by people with hypertension who shou devoid sodium.
- L 223 - abbreviations should be explained
- L 264- this modification should be given
Author Response
1.L19- small letters
Our response: Thank you for your suggestions. We have revised the manuscript and marked red color. Please look up the revised version of our manuscript.
- L41- small letters
Our response: Thank you for your suggestions. We have revised the manuscript and marked red color. Please look up the revised version of our manuscript.
- L69-all abbreviations should be explained
Our response: Thank you for your suggestions. We have revised the manuscript and marked red color. Please look up the revised version of our manuscript.
- L90-small letter
Our response: Thank you for your suggestions. We have revised the manuscript and marked red color. Please look up the revised version of our manuscript.
- L 90–why it issodium selenite? Sodium ion keep a water molecule, and cannot be consumed by people with hypertension who shoud evoid sodium.
Our response: Thank you for your suggestions. Because sodium selenite prevented diseases caused by selenium deficiency. Studies also showed that high concentration of sodium selenite could promote the proliferation of cell DNA and delay cell senescence. Moreover, the intake of sodium selenite could reduce the incidence of cancer, coronary heart disease, and hypertension. Therefore, sodium selenite was selected to prepare SeCS nanoparticles.
- L223 - abbreviations should be explained
Our response: Thank you for your suggestions. Since we have given full names for DPPH and ABTS in Lines 70 and 71, we use abbreviations here.
7.L 264- this modification should be given
Our response: Thank you for your suggestions. We have revised the manuscript and marked red color. Please look up the revised version of our manuscript.
Reviewer 2 Report
- Authors should provide different scales of SEM and TEM images.
- about XRD, in Figure -5, need more discussion in the text.
- Figure 6A, 7.4 pH should be included, in Figure 6B, Why 4 and 25oC only why not higher temperature 37oC, and higher temperature should check.
- Why at 25oC size increasing? need more discussion.
- Also, cell viability data will give more strengthen the manuscript.
Author Response
- Authors should provide different scales of SEM and TEM images.
Our response: Thank you for your suggestions. Transmission electron microscopy (TEM) was the best choice when you wanted to get information from the internal structure of samples, and scanning electron microscopy (SEM) was the first choice when you needed surface information of samples. Since the content of this study was mainly to observe the surface morphology of nanoparticles, we chose SEM instead of TEM. The results was shown in Figure 2. Moreover, different scales of SEM were added in Figure 2. Please look up the Figure 2.
- about XRD, in Figure -5, need more discussion in the text.
Our response: Thank you for your suggestions. We have revised the manuscript and marked red color. Please look up the revised version of our manuscript. (See Line 183-184)
- Figure 6A, 7.4 pH should be included, in Figure 6B, Why 4 and 25° C only why not higher temperature 37 ° C , and higher temperature should check.
Our response: Thank you for your suggestions. (1) On the one hand, pH-responsiveness was the most frequently investigated since pH values vary quite significantly in different tissues and cellular compartments.(Chiang et al.,Langmuir. 2012, 42, 15056−15064; Hua et al., Macromolecules. 2011, 44, 1298-1302; Liu et al., Langmuir. 2011,27, 3095-3099) The extracellular environment of a tumor had a lower pH (∼6.8) than blood and normal tissues (pH 7.4), whereas those of late endosome and lysosome were even lower (∼5.0− 5.5). (Gerweck et al., Cancer Res. 1996, 56, 1194−1198; Sun et al., Biomacromolecules. 2010, 4, 848−854; Gao, et al., 2010, 6, 1913−1920; Wang et al., RSC Adv. 2012, 2, 11953−11962; Hu et al., Mater. Chem. B. 2013, 1, 1109−1118) On the other hand, considering that the drug is given orally, it will pass through the stomach and small intestine of the human body, and under normal circumstances, the stomach and small intestine are acidic in the human body. For the above two reasons, we did not consider the neutral condition. (2) As materials are generally stored in cold storage or at room temperature, storage temperatures are 4 ° C and 25 ° C, while 37 ° C is the heating temperature, which is not suitable for storage. So temperatures of 37 degrees or higher are not considered.
- Why at 25oC size increasing? need more discussion.
Our response: It was possible that changes in the internal structure of the SeCS, such as a change in the crystal shape, led to a change in the amorphous state of SeCS to a crystalline state, resulting in nanoparticles cluster to make a increase in size. Moreover, we have added the reasons in the revised the manuscript and marked red color. Please look up the revised version of our manuscript.
- Also, cell viability data will give more strengthen the manuscript.
Our response: Thank you for your suggestion. The cell activity data you mentioned is what we need to study next, but we think it is inappropriate to add it to this article, because in this article, we mainly investigates the antioxidant activity of SeCS. Thank you very much!
Reviewer 3 Report
The authors prepared selenium-chondroitin-sulfate (SeCS) nanoparticles through an optimized chondroitin-sulfate adsorption to the surface of a selenium oxide. The nanoparticles were characterized by SEM, EDS, and FTIR, and an X-ray crystal structure was obtained. The selenium content of the particles was noted to be greater than that from other preparations. The pH stability and radical scavenging ability of the materials were investigated; however, the SeCS did not perform as required for potential biological materials.
The authors adequately optimized the preparation of SeCS nanoparticles with respect to reagent stoichiometry, temperature, and reaction time to reproducibly obtain average particle diameters of ~130 nm. The particles demonstrated antioxidant behavior in DPPH and ABTS radical scavenging assays, although the results were far inferior to the positive control ascorbic acid.
The manuscript requires careful editing for English grammar and usage prior to publication.
Statements that require clarification or correction include:
- Line 20 and line 292 What is the “redox reaction method”. There should be a reference.
- Line 35 Remove “As is known to all”, since nothing is known to everyone especially as it relates to chemistry.
- Line 42 state the “extremely narrow margin” as an actual range of numbers.
- Line 45 Nanotechnology is not a “new technology”. It was reported over 50 years ago by Taniguchi.
- Line 51 SeNPs “possess higher bioactivity” in what regard, specifically? Toxicity?
- Line 64 “using CS as a stabilizer to stable SeNPs” is confusing. Why would a stabilizer be necessary if the SeNPs are already stable?
- Line 83 “occurred to aggregation” should be changed to “aggregated”.
- Line 83 what is meant by, “reduce its hydrophilic group’ function”?
- Line 88 “Different characters indicate significant different between different groups” is nonsensical. That sentence must be clarified. What differences specifically? What constitutes a group?
- Line 100 “made the particle size became larger due to the unstable reaction system” is unclear. In what sense is the reaction system unstable? Based on what experimental evidence?
- Line 105 “was considered as” should be changed to ‘has’.
- Line 128 “screened out the best” should be changed to ‘optimized’.
- Line 161 “is used to ascertain the formation mechanism of SeCS” Is incorrect as written. The word “mechanism” should be removed.
- Line 194 “This was probably attributed to the strongest electrostatic interaction between CS and SeNPs at pH 8.0 due to the pH sensitivity of CS.” Perhaps “between anionic CS” and “due to the sensitivity of CS at low pH should be added for clarification.
- Line 204 “significant different between different groups” is meaningless. What differences specifically?
- Line 215 “particularly in the ABTS scavenging assay” should be added to the last sentence of the paragraph.
- Line 298 The concluding sentence is overstated. It should be changed to “the higher thermal stability and the improved antioxidant properties of SeNPs capped with CS compared to other SeNPs.”
In summary, the described nanoparticles offer slightly improved properties over other selenium-based nanoparticles; however, their performance still promises little practical application due to limited pH stability and moderate antioxidant activity. Even so, results of this study could spur innovation in the formulation of improved future selenium-based nanoparticles. As such, I am in favor of publication after the authors adequately address the points listed above and correct the manuscript for English grammar and usage.
Author Response
- Line 20 and line 292 What is the “redox reaction method”. There should be a reference.
Our response: Because the reaction between ascorbic acid (VC) and sodium selenite (Na2SeO3) is redox reaction, and in this reaction, VC is reduing agent. Thus, in Line 20, we added relevant information in the revised version of manuscript and marked red color. Meanwhile, in Line 292, we also revised the manuscript and marked red color. Please look up the revised version of our manuscript.
- Line 35 Remove “As is known to all”, since nothing is known to everyone especially as it relates to chemistry.
Our response: Thank you for your suggestions. We have deleted “As is known to all” and revised the manuscript and marked red color. Please look up the revised version of our manuscript.
- Line 42 state the “extremely narrow margin” as an actual range of numbers.
Our response: It is difficult to describe the precise range. Because for different people, the amount of selenium intake is different. Moreover, the selenium supplement standard is not uniform in different place. For example, the World Health Organization (WHO) recommends a daily selenium intake of 50-200 μg/d for healthy adults, with a maximum tolerable intake of 400 μg/d, without gender differences. However, in 2014, the European Food Safety Authority (EFSA) recommended a dietary allowance of Se(70 μg/d) for adults. Moreover, studies showed that a 300 μg/d dose of selenium taken for 5 years increased mortality 10 years later. Thus, 300 μg/d dose or high-dose selenium supplements should be avoided.(Ali Razaghi et al., European Journal of Cancer, 2021, 155, 256-267. ;Rayman et al., Free Radical Biology and Medicine, 2018, 127, 46-54.). Whereas, for cancer patients, studies have shown that selenium supplements can be as high as 400 μg/d. Therefore, it's hard to use a number to describe this narrow margin.
- Line 45 Nanotechnology is not a “new technology”. It was reported over 50 years ago by Taniguchi.
Our response: We have changed “ new technology” to “technology” and marked red color. Please look up the revised version of our manuscript.
- Line 51 SeNPs “possess higher bioactivity” in what regard, specifically? Toxicity?
Our response: We have added the bioactivity in the revised version of the manuscript and marked red color. Please look up the revised version of our manuscript.
- Line 64 “using CS as a stabilizer to stable SeNPs” is confusing. Why would a stabilizer be necessary if the SeNPs are already stable?
Our response: Because SeNPs has high specific surface energy, it is very unstable and easy to accumulate and form precipitation. Therefore, CS should be added as a stabilizer to avoid SeNPs aggregation, so as to improve the stability of SeNPs.
- Line 83 “occurred to aggregation” should be changed to “aggregated”.
Our response: Thank you for your suggestions. We have revised the manuscript and marked red color. Please look up the revised version of our manuscript.
- Line 83 what is meant by, “reduce its hydrophilic group’ function”?
Our response: CS has abundant hydrophilic groups, such as hydroxyl and carboxylic acid groups. When CS occurs to aggregation, these hydrophilic groups in CS molecules can promote the formation of intermolecular hydrogen bonds in polymer chains. Thus, the electrostatic interaction between hydrophilic groups in CS molecules and SeNPs is reduced. Therefore, “reduce its hydrophilic group’function’’ means that CS molecules aggregated to reduce the electrostatic interaction between hydrophilic groups in CS molecules and SeNPs.
- Line 88 “Different characters indicate significant different between different groups” is nonsensical. That sentence must be clarified. What differences specifically? What constitutes a group?
Our response: Thank you for your suggestions. We have revised the manuscript and marked red color. Please look up the revised version of our manuscript.
- Line 100 “made the particle size became largerdue to the unstable reaction system” is unclear. In what sense is the reaction system unstable? Based on what experimental evidence?
Our response: Because when the molar ratio of Na2SeO3 to Vc is greater than 1:8, excess VC will bind to the surface of the original selenium nucleus, resulting in the aggregation of the selenium nucleus and eventually the aggregation of SeCS to make the particle size became larger. This change in morphology is accompanied by a change in the color of the reaction solution.
- Line 105 “was considered as” should be changed to ‘has’.
Our response: Thank you for your suggestions. We have changed “was considered as” to “had” in the revised version of the manuscript and marked red color. Please look up the revised version of our manuscript.
- Line 128 “screened out the best” should be changed to ‘optimized’.
Our response: Thank you for your suggestions. We have changed “screened out the best” to “optimized” in the revised version of the manuscript and marked red color. Please look up the revised version of our manuscript.
- Line 161 “is used to ascertain the formation mechanism of SeCS” Is incorrect as written. The word “mechanism” should be removed.
Our response: Thank you for your suggestions. We have deleted the word “mechanism” in the revised version of the manuscript and marked red color. Please look up the revised version of our manuscript.
- Line 194 “This was probably attributed to the strongest electrostatic interaction between CS and SeNPs at pH 8.0 due to the pH sensitivity of CS.” Perhaps “between anionicCS” and “due to the sensitivity of CS at low pH should be added for clarification.
Our response: Thank you for your suggestions. We have revised the manuscript and marked red color. Please look up the revised version of our manuscript.
- Line 204 “significant different between different groups” is meaningless. What differences specifically?
Our response: Thank you for your suggestions. We have revised the manuscript and marked red color. Please look up the revised version of our manuscript.
- Line 215 “particularly in the ABTS scavenging assay”should be added to the last sentence of the paragraph.
Our response: Thank you for your suggestions. We have added “particularly in the ABTS scavenging assay” in the revised version of the manuscript and marked red color. Please look up the revised version of our manuscript.
- Line 298 The concluding sentence is overstated. It should be changed to “the higher thermal stability and the improved antioxidant properties of SeNPs capped with CS compared to other SeNPs.”
Our response: The stability in this study refers to pH and storage stability, so it is not suitable to be described as thermal stability. Thus, we have changed to “the higher stability and the improved antioxidant properties of SeNPs capped with CS compared to SeNPs” .
Reviewer 4 Report
- The manuscript requires improve presentation and clarity. English proofreading is required. Some typo and grammar errors must be avoided.
- The size of particles is in the borderline to be considered nanoparticles, please add reference(s) to support the characteristics of the material in this category.
- The relevance of marine sources should be reinforced. Differences among CS sources should be mentioned. The features of CS-from shark should be clearly sentenced.
- The Catalog Number of the sigma product should be added to methods section. The comparisson among composition of other CS is desirable. (Please take a look: https://doi.org/10.1016/j.carbpol.2015.08.006 https://www.mdpi.com/1420-3049/20/3/4277/pdf)
- Legends for figures should be clear. In the fig 2-45the legends could include relevant details. In the figures 1, 6 and 7 they should include clarity regarding the meaning of characters. Moreover, in the latter, the letters should mark just the differences, but not all the columns with a letter as subgroup. Why figure 4 has not the SeNPs spectrum?
- The presentation of results in the same order in the figures is desirable. Moreover, probably CS then, SeNp, and finally SeCs, should be considered in all figures.
- Discussion should highlights the putative advantages of the proposed particles considering similar nanoparticles recently reported.
Author Response
- The manuscript requires improve presentation and clarity. English proofreading is required. Some typo and grammar errors must be avoided.
Our response: We have revised some typo and grammar mistakes in the revised version of the manuscript and marked red color. Please look up the revised version of manuscript.
- The size of particles is in the borderline to be considered nanoparticles, please add reference(s) to support the characteristics of the material in this category.
Our response: According to a previous report, nanoparticles with a diameter <200 nm would accumulate at tumor sites (Shanthi, et al., RSC advance, 2015, 5(56), 44998-45014; Tang et al., Journal of Functional Foods, 2021, 78,104359 ). In this study, the size of SeCS was much smaller than 200nm. Thus, the SeCS possessed nanobiological effects.
- The relevance of marine sources should be reinforced. Differences among CS sources should be mentioned. The features of CS-from shark should be clearly sentenced.
Our response: Thank you for your suggestions. We have revised the introduction and method section in the manuscript and marked red color. Please look up the revised version of manuscript.
- The Catalog Number of the sigma product should be added to methods section. The comparisson among composition of other CS is desirable. (Please take a look:https://doi.org/1016/j.carbpol.2015.08.006;https://www.mdpi.com/1420-3049/20/3/4277/pdf)
Our response: Thank you for your suggestions. First, I am sorry for the writing error in the manuscript. The CS is purchased from Aladdin. And the molecular weight of CS is 499.37. The Catalog Number of the aladdin product is C107703. We have revised the method section in this manuscript and marked red color. Please look up the revised version of manuscript.
- Legends for figures should be clear. In the fig 2-45the legends could include relevant details. In the figures 1, 6 and 7 they should include clarity regarding the meaning of characters. Moreover, in the latter, the letters should mark just the differences, but not all the columns with a letter as subgroup. Why figure 4 has not the SeNPs spectrum?
Our response: Thank you for your suggestions. We have revised the all figures legends. Moreover, we added the SeNPs spectrum in Figure 4. Please look up the revised version of manuscript.
- The presentation of results in the same order in the figures is desirable. Moreover, probably CS then, SeNp, and finally SeCs, should be considered in all figures.
Our response: Thank you for your suggestions. We have revised the figures 1-2 and 4-7 in the revised version of manuscript. Please look up the revised manuscript.
- Discussion should highlights the putative advantages of the proposed particles considering similar nanoparticles recently reported.
Our response: In the discussion section, we mentioned that the SeCS prepared in this study contained selenium up to 33.18%, which is higher than that reported in recently studies (Gao, et al. Int. J. Biol. Macromol. 2020, 142: 265-276 ). This is a huge advantage because higher selenium levels help to enhance the biological activity of nanoparticles.
Round 2
Reviewer 2 Report
Recommended for publication.
Author Response
1. Recommendation for publication.
Our response: Thank you for your recommendation.